# Using Low-Cost Air Quality Sensor Networks to Improve the Spatial and Temporal Resolution of Concentration Maps

**DOI:** 10.3390/ijerph16071252

**Published:** 2019-04-08

**Authors:** Faraz Enayati Ahangar, Frank R. Freedman, Akula Venkatram

**Affiliations:** 1Department of Mechanical Engineering, University of California, Riverside, CA 92521, USA; fenay001@ucr.edu; 2Department of Meteorology and Climate Science, San Jose State University, San Jose, CA 95192, USA; frank.freedman@sjsu.edu

**Keywords:** LCAQM, dispersion modeling, spatial interpolation, Kriging, imperial valley, PM_2.5_

## Abstract

We present an approach to analyzing fine particulate matter (PM_2.5_) data from a network of “low cost air quality monitors” (LCAQM) to obtain a finely resolved concentration map. In the approach, based on a dispersion model, we first identify the probable locations of the sources, and then estimate the magnitudes of the emissions from these sources by fitting model estimates of concentrations to corresponding measurements. The emissions are then used to estimate concentrations on a grid covering the domain of interest. The residuals between model estimates at the monitor locations and the measured concentrations are then interpolated to the grid points using Kriging. We illustrate this approach by applying it to a network of 20 LCAQMs located in the Imperial Valley of Southern California. Estimating the underlying mean concentration field with a dispersion model provides a more realistic estimate of the spatial distribution of PM_2.5_ concentrations than that from the Kriging observations directly.

## 1. Introduction

Several studies indicate that exposure to fine particulate matter (PM_2.5_) concentrations is associated with several adverse health oucomes that include cardiovascular diseases [1], lung cancer, cardiopulmonary mortality, and asthma [2,3]. These results have motivated the expansion of programs to measure PM_2.5_ concentrations in several affected communities.

Most of the monitoring networks currently maintained by state and federal agencies do not provide the information at the spatial and temporal resolution required to assess the impact of pollution sources on health. For example, assessing the impact of vehicle emissions on the health of people living next to highways requires spatial resolution of tens of meters. Because expansion of networks with currently accepted instrumentation is expensive, agencies are considering monitors that are relatively inexpensive, and hence referred to as Low-Cost Air Quality Monitors (LCAQM). Recent development in sensor technology has improved the performance of these sensors [4,5]. These monitors can expand the capability of current networks if they are calibrated against agency monitors at regular intervals [6,7,8]. The data gathered by these sensors can enhance the information provided by traditional networks, in particular by helping to detect local hot spots in concentration patterns. This will provide significantly improved information for air quality management purposes [9,10,11].

The expansion and availability of wireless networks have made the data captured by LCAQMs available in real-time, which makes these types of monitoring ideal for community monitoring. Several communities have already installed networks of LCAQMs to gauge their usefulness. Mead et al. [6] deployed a dense LCAQM network that was used to assess air pollution exposure at a fine scale. Johnson et al. [12] developed an LCAQM system using several different low-cost particulate matter (PM) sensors to assess concentration patterns on a city scale. They showed that the correlation between the measurments from the sensors and those from reference monitors were good. Similar networks have also been used inside buildings to help monitor and enhance indoor air quality [13,14].

Although LCAQMs provide better spatial coverage than that from reference monitors, it is still necessary to fill in the gaps between these sensors using some method. Land Use Regression (LUR) is one such method, which achieves this objective by relating measurements to land use factors through linear models [15,16,17]. LUR models are popular because (1) they are anchored to observed concentration patterns, (2) they are based on readily accessible land use variables, and (3) they do not require explicit modeling of the processes that govern the concentration field. Once the model has been calibrated with measurements, it can be used to estimate concentrations at a location of interest using local land use factors as inputs. LUR models have been used to estimate temporal changes in concentration patterns by including time varying meteorological variables as predictors or recalibrating the model based on temporal changes in concentrations observed at fixed monitors [18,19].

The main limitation of LUR models is that they do not relate emissions to concentrations, although the relationship is implicitly accounted for in the land use variables. Thus, the LUR model provides limited guidance on mitigating air pollution through emission control. Here, we show how dispersion models can be used to overcome this disadvantage of the LUR model and at the same time provide the concentration maps tied to observations. Because dispersion models explicitly relate source strength to air quality, they can be used to predict changes in the concentration field when emissions change [20] (Hoogh et al., 2014). Furthermore, dispersion models provide the mechanistic foundation required to relate the temporal variation in meteorology to the corresponding variation in concentration patterns.

Starting in 2016, the Identifying Violations Affecting Neighborhoods (IVAN) LCAQM network has been operating, developed as a NIH-funded collaboration of community, academic, nongovernmental and governmental partners designed to augment routine state agency particulate measurements in the Imperial Valley. The network consists of 40 modified Dylos monitors over the valley to address various air quality concerns of the community [21]. The relatively high density of the network provides an opportunity to illustrate our approach to applying a dispersion model to increase the spatial and temporal coverage of the network. In this paper, we analyze data from the IVAN network to (1) improve the spatial resolution of the LCAQM network, and (2) estimate the source–receptor relationships embedded in the observations. The approach consists of using the model to estimate the unknown emissions that provide the best fit between model estimates and corresponding LCAQM measurements. The residuals between the model estimates and measurements are then used to create a spatial concentration map at the desired resolution, where the model estimates provide the underlying trend. This study differs from previous studies [22,23,24], in which a dispersion model was used to construct concentration maps, in that the model is used to estimate emissions from the measured concentration field. The estimated emissions are then used to model the concentration patterns.

## 2. Methodology

### 2.1. Sensor and Source Locations

The Imperial Valley is located in Southern California in Imperial County. It is roughly 80 km from north to south and 40 km from east to west. It is bordered by the Colorado River to the east, the Salton Sea to the northwest, the US–Mexico Border (Mexicali) to the south, by desert topographic features to the west and east. Most of the land in the valley is devoted to agricultural activities. According to the California Air Resources Board (CARB), most of the PM emissions in the valley originate from unpaved roads and fugitive windblown dust within the valley [25]. Windblown dust from the desert in the west also makes a contribution to the PM_2.5_ concentrations. Emissions from Mexicali have an impact on concentrations in the southern part of the valley. Historically, PM_10_ and PM_2.5_ concentrations at agency monitors across the county have exceeded federal and state air quality standards, and the area is currently non-attainment for both PM_10_ and PM_2.5_ (https://www.arb.ca.gov/desig/adm/adm.htm). In terms of health impacts, Imperial County has the highest rates of asthma emergency room visits and hospitalization among school-age children of all counties in California (http://www.cehtp.org/page/main).

The regulatory air monitoring system in the valley consists of five FEM stations that measure different gaseous and particle pollutants. Three of these stations measure daily-averaged PM_2.5_. The IVAN network of low-cost sensors was developed in response to the community’s concerns about the air quality and its desire for more local concentration data [26]. This is one of the largest community-based air monitoring networks in the US and is considered the first community-designed network of its size in the world. The network uses modified light-scattering particle counters, DC 1700, manufactured by Dylos Corporation. The particle counter provides four size bin measurements that can be accessed in real-time through the internet (>0.5 µm, >1.0 µm, >2.5 µm, and >10 µm) [27].

The data, which is displayed online through a website [28], show that, with relative humidity corrections, the PM_2.5_ concentrations from the Dylos instruments can be related to measurements from multiple reference instruments. These sensors were operational during most of 2017 (Figure 1). The locations of these sensors were chosen by local community members and the staff of California regulatory agencies to address concerns about sensitive receptors, the possibility of a flag notification system, and real-time representation of air quality in populated areas [26].

To check the validity of the measurements, we compared the data measured by Dylos devices in 2017 to one of the regulatory stations in the area. Figure 2 shows the comparison of ${\mathrm{PM}}_{2.5}$ daily averaged values measured at the state agency Calexico-Ethel Street station with the daily averaged measurement from the closest Dylos sensor. The Dylos measurements were well correlated with the observations from the fixed monitor. Another comparison of sensor data with FEM measurement has also showed similar results [27]. Note that at concentrations below 15 $\mu g/m^{3},$ there is considerable scatter between the two measurements and the Dylos measurements are about 20% lower than those from the CARB station. This bias is much lower at higher concentrations. Our modeling analysis is based on monthly and annually averaged concentrations to reduce the impact of the scatter.

### 2.2. Modeling Approach

A main source of PM in the valley is the valley itself, which includes sources such as unpaved roads, dust from agricultural activity on farmlands, and traffic. Particulate matter can also be transported into the valley from various sources, including dust from the surrounding deserts, exposed lake bed from the Salton Sea, and emissions originating from activities in Mexico (Figure 3).

We use the following the steps to apply the dispersion model: (1) specify sources through their locations and geometry and assign unit emissions to them. Boundaries are treated as line sources and the entire valley is modeled as an area source. The ${\mathrm{PM}}_{2.5}$ emissions originated from this area source come from various sources, including point, area, and on-road mobile sources. Area sources, which consist of fugitive dust and unpaved road emissions, are estimated to account for 83 percent of the total emissions [25]; (2) construct meteorological inputs for the dispersion model using routine meteorological measurements from an airport located in the valley; (3) run dispersion model with unit emissions for the sources and regress model results on measured concentrations to estimate emissions from sources; and (4) use estimated emissions and meteorological inputs to estimate concentrations at locations of interest.

We use 11 straight lines, laid along the borders of the valley, to represent different source regions that can contribute to the ${\mathrm{PM}}_{2.5}$ concentrations in the valley. The line sources on the west side of the valley represent the desert, which is referred to as the West Desert. The southern edge of the Salton Sea is represented as a separate line source. Line sources on the east represent the East Desert. Emissions from these sources are primarily fugitive windblown dust. The line sources on the south side represent anthropogenic emissions and windblown dust from Mexico.

Hourly concentrations resulting from the line source are modeled using the approach in R-LINE [29]. The concentration field is computed using an approximation to the integral that corresponds to the contributions of point sources along the length of a finite line source when the wind direction is at an arbitrary angle relative to the line. The horizontal and vertical spreads of plumes originating from the line source are computed using functions that depend on surface micrometeorology and downwind distance [30].

Hourly concentrations from the area source is modeled using the approach in AERMOD [31] in which the area integral representing the contributions from sources upwind of a receptor is evaluated with a set of line sources perpendicular to the wind; the number of line sources is determined by the convergence criterion used to evaluate the area integral. Because these models are steady state models that use straight line trajectories, the source–receptor relationships embodied in them are not likely to be accurate over spatial scales exceeding 10 km. Their application implies that the primary contribution to PM concentrations at a receptor originate from sources within this distance.

The concentrations from each of the 11 sources were calculated for at each IVAN monitor location using a unit emission rate. Then the concentrations from 11 sources were grouped to different source categories: West Desert ($T_{W}$), Salton Sea ($T_{S}$), East Desert ($T_{E}$), Mexico ($T_{M}$), and Valley ($T_{V}$). The emissions from each of these categories were then computed as the non-negative regression coefficients of the following linear equation that provides the best fit to the observed concentrations, $C_{oi}$, in the least squares sense.
(1)$$C_{oi}=E_{w}T_{Wi}+E_{s}T_{Si}+E_{E}T_{Ei}+E_{M}T_{Mi}+E_{V}T_{Vi}+\epsilon_{i}$$
where *i* corresponds to the location of the Dylos sensor, $E_{W}$, $E_{S}$, $E_{E}$, $E_{M}$, and $E_{V}$ are the total emission rates of West Desert, Salton Sea, East Desert, Mexico, and Valley, respectively, and $\epsilon_{i}$ is the residual at the receptor.

We use bootstrapping to estimate the 95% confidence intervals for the estimated emissions. The residuals between model estimates and observed ${\mathrm{PM}}_{2.5}$ concentrations are added randomly to model estimates at each receptor to create pseudo-observations, which are then fitted to model estimates to create a distribution of emissions. The 95% confidence interval corresponds to the 2.5 and 97.5 percentiles of the resulting distributions.

### 2.3. Meteorological Inputs

Hourly meteorological inputs for the model were derived using the AERMET [31] processor from routine measurements made at the Imperial County Airport Meteorological station (KIPL) near El Centro. These data show that the wind is predominantly from the west and southwest during the year (Figure 4). High winds are common during the daytime and low winds usually occur during the night. Most of the low wind speeds are westerly. Representing the meteorology over the large area of the valley with measurements from a single station introduces uncertainty in the modeling exercise; it can be improved by using data from multiple stations or outputs from a meteorological model.

### 2.4. Concentration Measurement

Measurements from 36 IVAN stations were available during 2017. The sensors that were not operating for at least six months of the year were not used in the analysis. Moreover, some of the sensors that measured high concentration values that were inconsistent with the concentration levels in Calexico were also removed from the analysis. Figure 5 shows the locations of sensors that were operational for more than 75% of the year, together with the corresponding measured annually averaged concentrations. Typically, the highest concentrations occur in the southern part of the valley, which is consistent with results from related studies [25,32] 

## 3. Results and Discussion

### 3.1. Modeling Results

We first considered annually averaged ${\mathrm{PM}}_{2.5}$ concentrations in estimating emissions. The model was fitted to data from a sensor only if it was operational for at least six months during 2017. This filtering resulted in 21 sensors being considered in the analysis. Only hours with friction velocities above 0.1 m/s were considered to avoid conditions that are considered by AERMET to be calm conditions.

We estimated the emission rates using both annually averaged and monthly averaged PM_2.5_ observations. The modeled concentration from each source was determined for each hour and then averaged over the averaging period. The averaged model concentrations were then fitted to observed values over the corresponding averaging period to estimate the emission rates from the prescribed sources. The differences between the emission values based on monthly and annually averaged concentrations provides information on the uncertainty in the emission estimates.

Figure 6a shows model performance using annually averaged observations. The performance of the model is described using two measures: The fraction of observations within a factor of the model estimates, and the geometric standard deviation of the ratio of the observed-to-modeled concentration [33]. All of the model estimates fall within a factor of two of the observations, and the geometric standard deviation is $s_{g}$ = 1.15.

Figure 6b shows the estimated emission rates from the different source categories. These results indicate that the majority of the PM_2.5_ emissions that affect PM_2.5_ concentrations at receptors considered in this study originate from within the Valley during the year. Note that the contributions of these emissions to PM_2.5_ concentrations at a specific receptor depends on the source-receptor relationship that governs the receptor. For example, the PM_2.5_ concentrations at Calexico are dominated by the emissions from Mexico when the winds are from the south. 

Our estimate of 12.4 tons/day from the valley is surprisingly close to the bottom-up emission estimate of 14.1 tons per day made by the California Air Resources Board [25]. The Salton Sea emits 3.5 tons per day while the West Desert emission is 2.8 tons per day. These results indicate that, on an annual basis, the observed ${\mathrm{PM}}_{2.5}$ concentrations are dominated by local sources. The East Desert has the lowest emission contribution, emitting 0.34 tons per day for the year. This is mainly because the dominant wind direction is westerly, resulting in transport of emissions from Salton Sea and West Desert more often than from the east. Mexico has also a significant contribution of 2.8 tons per day.

Table 1 shows the range of emission rates for different sources. The estimated uncertainty ranges from 0.5 of the mean emission estimate in the Valley to 8.3 times the estimated mean emission from East Desert.

Figure 7a shows the performance of the model in describing monthly averaged observations: 92 percent of model estimates fall within a factor of two of the observed values.

Figure 7b shows the estimated emission rates from the different source categories assuming that these emission rates do not vary from month to month. The mean emission estimates derived from monthly averaged concentrations are similar to those from the annually averaged concentrations. The Valley still has the highest emission rates with 11.4 tons per day, the Salton Sea emits 3.2 tons per day and west desert emits 2.1 tons per day. The estimate for Mexico is 3.2 tons per day and is higher than the previous estimates from the model. East Desert still has the lowest emission rate, 1.9 tons per day, but higher than the previous estimates. Bootstrapping estimates of the 95% confidence intervals of emission rates derived from monthly averages, shown in Table 2, are consistent with those obtained from annually averaged data.

Figure 8 shows the performance of the model for four sensors at different locations assuming constant emission rates of Table 2 throughout the year. Most of the values fall within a factor of two of the observations.

Figure 9 compares model estimates of monthly averaged PM_2.5_ compared with corresponding data from the FRM station at Calexico-Ethel Street. The measured values are consistent with the model estimates even though these measurements were not used in the analysis.

We estimated the variation of emission rates by month by fitting model concentration estimates to monthly averaged measured concentrations for each month at the 21 receptors. We only considered sensors with at least 100 hours of measurements each month. Table 3 shows the model performance.

All of the observations fall within a factor of two of the observations for all of the months. The coefficient of determination, *r*^2^, is as high as 0.55 (for January and May), while low values occur for a few months. Low *r*^2^ values are usually associated with small variances in the observed concentrations. The largest concentration variance occurs in December when the concentration varies between 7 $\mu /{\mathrm{gm}}^{3}$ and 25 $\mu /{\mathrm{gm}}^{3}$. Moreover, this month has the highest concentration levels.

Table 3 also shows the emission rates for different months of the year calculated by the model. The Valley is the main source of particulate matters during the year. The largest emissions come during December (27 tons per day from the Valley) when the highest concentrations are observed. East Desert usually has the lowest values during the year with a large variation of 0 to 7.6 tons per day, depending on the wind direction and speed. The Salton Sea is also one of the major sources of emissions, especially during June (9.5 tons per day) and August (8.7 tons per day). Mexico emits large amounts of emissions during December (10.5 tons per day). The highest contribution of the West Desert occurs during May, when it is 5.4 tons per day.

### 3.2. Sensitivity of the Model to $\sigma_{z0}$

Here we look at the sensitivity of modeled annual emissions to the initial vertical plume spread, $\sigma_{z0}$, used to represent the upwind extent of the line sources on the border. This value is assumed to be 10 m in the runs corresponding to the previous results. Figure 5, Figure 6, Figure 7, Figure 8 and Figure 9 shows the change in emission rates with different initial plume spreads. Increasing the initial plume height reduces the emission rates coming from the valley and increases the emission rates from other sources (Figure 10). This is expected since increasing the initial plume spread increases the dispersion from the sources around the valley and decreases the concentrations. This results in higher emission rates for these sources to account for increased dispersion.

These results indicate that the emissions estimated by fitting model estimates to corresponding measurements depend on the parameters of the dispersion model. However, even with their uncertainty, the results still provide information that can be used to mitigate the PM_2.5_ problem in the Valley. The results consistently indicate that most of the emissions in the area originate from the Valley for different vertical plume spreads. The East Desert also contributes the least to the pollution, regardless of the initial vertical plume spread.

### 3.3. Using Residual Kriging to Improve Concentration Maps

As discussed in Section 1, LUR is commonly used to provide spatially continuous concentration fields required for exposure assessment. As an alternative that offers advantages mentioned in that section, we propose a method that combines dispersion modeling with Kriging to construct spatially continuous concentration fields [23,24]. Kriging has been used commonly to interpolate observations at different locations and create maps in a variety of applications [34]. Simple Kriging includes calculating the concentration at each point by assuming that the observed concentration is the sum of a spatially constant mean and a local fluctuation which is assumed to be spatially isotropic and homogeneous. This assumption is not generally valid in local-scale air pollution applications since the mean concentration at a receptor is governed by the spatial distribution of emissions relative to the receptor and associated meteorology. We can improve upon simple Kriging by making the more reasonable assumption that the mean concentration field is best estimated with a dispersion model that accounts for the emission distribution and meteorology. Once this mean field is determined by the dispersion model, the residuals between model estimates and observations at receptors are then more likely to be consistent with the homogeneity and isotropy assumptions of simple Kriging. This approach has been applied to produce concentration maps for different pollutants [22,24]. We use the following steps to produce the residual kriging concentration maps presented here:Use the dispersion model with fitted emissions to estimate concentrations at monitoring stations and calculate residuals between model estimates and observations.Use kriging to construct a field of residuals over a grid at a specified spatial resolution over the study domain.Estimate concentrations using the dispersion model with fitted emissions over the grid points of the study domain.Add the residuals computed from step 2 to the model estimates from step 3 to create concentration maps.

Figure 11, constructed using a grid resolution of 1000 m, shows that there are positive residuals (model overestimations) in the southwest valley next to the West Desert and U.S. Mexican border, and the northern portion of the valley. Areas of negative residuals (model underestimations) are located in areas spanning the center of the valley. We do not have the information to identify their precise causes, but we can suggest that they are related to actual local emissions being different than the source category averages used to model the concentrations. Temporal variations of emissions on hourly and daily scales, not accounted for in the modeling, could also play a role.

Figure 12a shows the concentration map based on simple Kriging. The map shows reasonable spatial variation at the regions with a higher density of sensors, for example in the southern valley (see Figure 5 for monitor locations) However, in the regions close to emission sources like the West Desert, where there are a very small number of sensors, there is little variation in ${\mathrm{PM}}_{2.5}$ concentrations. Figure 12b shows the concentration map produced with residual Kriging. The concentration pattern now reflects the mean concentration field determined by the dispersion model based on inferred emissions, and shows greater spatial variation than that from simple Kriging. This is emphasized in the histogram of Figure 13. Simple Kriging produces concentrations close to the overall mean field, while residual Kriging reflects a more realistic variation corresponding to the mean concentration field estimated by the dispersion model.

To evaluate the results from residual kriging, we use a leave-one-out cross-validation technique. We leave out a location from the observations and compare the interpolated value with the observed value. We expect the interpolation results from residual Kriging to improve upon those from simple Kriging, especially at locations with a low density of sensors. Figure 14 shows the interpolation results for simple and residual Kriging compared to each other at different locations. Residual kriging improves the coefficient of determination, *r*^2^, from 0.05 to 0.31 at Westmorland, 0.57 to 0.62 at Seeley, and 0.77 to 0.92 at Holtville compared to simple kriging. 

## 4. Summary and Conclusions

This paper presents an approach to analyzing PM_2.5_ data from a network of “low cost air quality monitors” (LCAQMs) that provides much better spatial resolution than that from the sparser network of FRM/FEM monitors. We illustrate this approach by applying it to a network of LCAQMs located in the Imperial Valley of southern California. The sensors measured hourly ${\mathrm{PM}}_{2.5}$ concentrations for over 20 locations across the valley during 2017.

We show how a dispersion model can be used to estimate the primary PM_2.5_ emissions that contribute to concentrations at the monitors: The emissions correspond to the best fit between model estimates and measurements at the monitors. The uncertainty in these emissions is computed using a bootstrapping technique.

Our interpretation of the data indicates that current annually averaged PM_2.5_ concentrations within the Imperial Valley are dominated by the emissions originating from the valley, which have an average emission rate of 11 to 13 tons per day. This is consistent with the earlier estimates from CARB of 14.1 tons per day [25] (CARB, 2018). Most of the remaining emissions originate from the Salton Sea, Mexico, and West Desert. The desert on the east of the Valley has the lowest contribution to the concentrations mainly because of prevailing westerly winds.

These emissions in combination with the governing meteorology are then used to estimate concentrations on a continuous grid that has a much finer spatial resolution than that of the LCAQMs. The residuals between model estimates at the monitor locations and the measured concentrations are then interpolated using Kriging to create highly resolved PM_2.5_ maps, a method we term “residual Kriging. We show that this incorporation of the underlying processes of determining the mean concentration field, through a dispersion model, provides a more realistic estimate of the spatial distribution of PM_2.5_ concentrations than that from a purely statistical technique, such as simple Kriging, which interpolates concentration values themselves rather than model versus measured residuals

Further improvement of the emission determination, for example by explicitly modeling known pollution sources rather than broad source areas and categories as done in this study, can improve the results from the dispersion model. Also, a more uniform LCAQM network which covers most of the area of the interest, and not just more populated areas, can help to provide better air pollution maps and quantify emission rates from potential sources more accurately.

## Figures and Tables

**Figure 1 ijerph-16-01252-f001:**
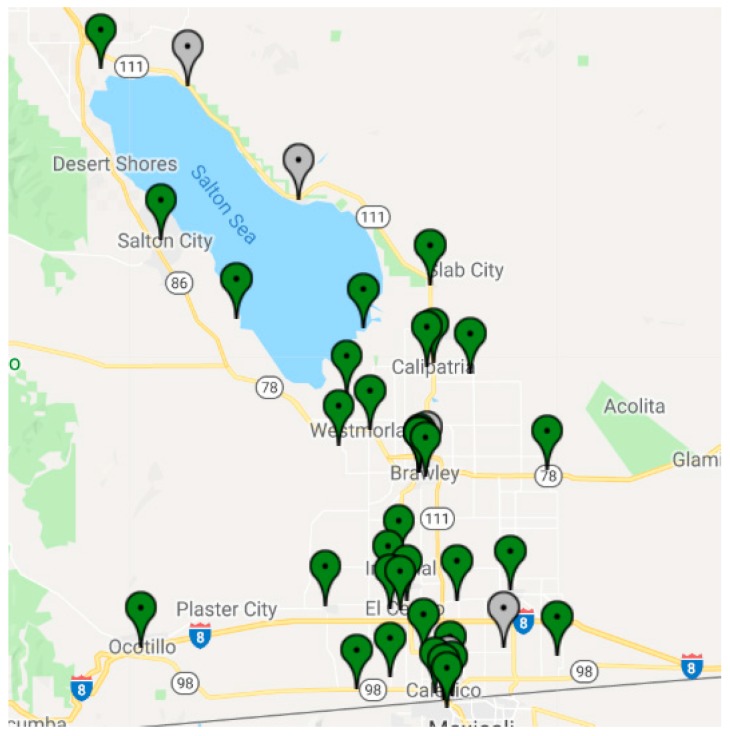
Map of IVAN low-cost sensors in the Imperial Valley for a particular day [28]. Green pins are monitors operating on the day, grey pins are monitors that were offline.

**Figure 2 ijerph-16-01252-f002:**
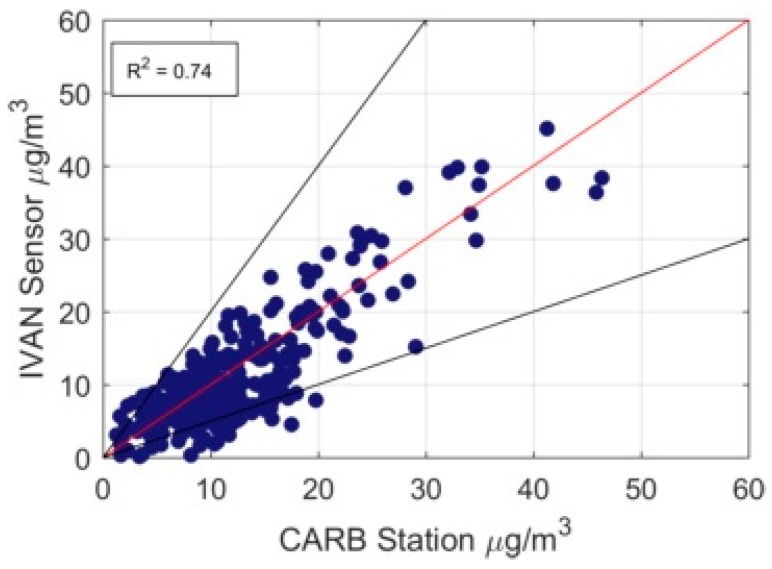
Daily averaged concentrations over the year 2017 of the IVAN sensor in Calexico compared with the regulatory agency Calexico-Ethel St. street station.

**Figure 3 ijerph-16-01252-f003:**
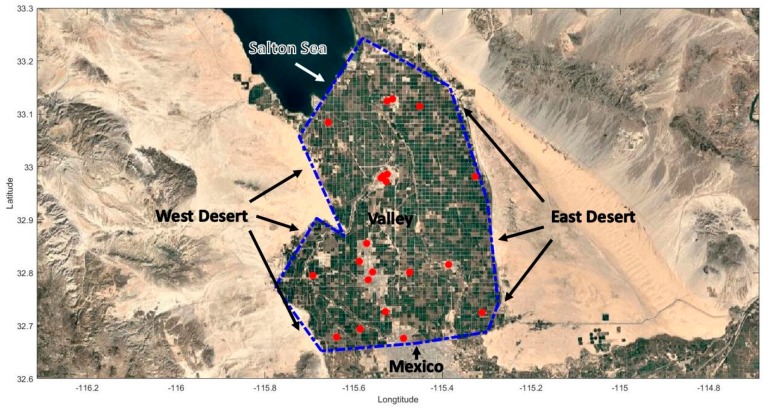
Locations of sources (blue dashed lines) and IVAN air monitors (red dots) used in dispersion modeling.

**Figure 4 ijerph-16-01252-f004:**
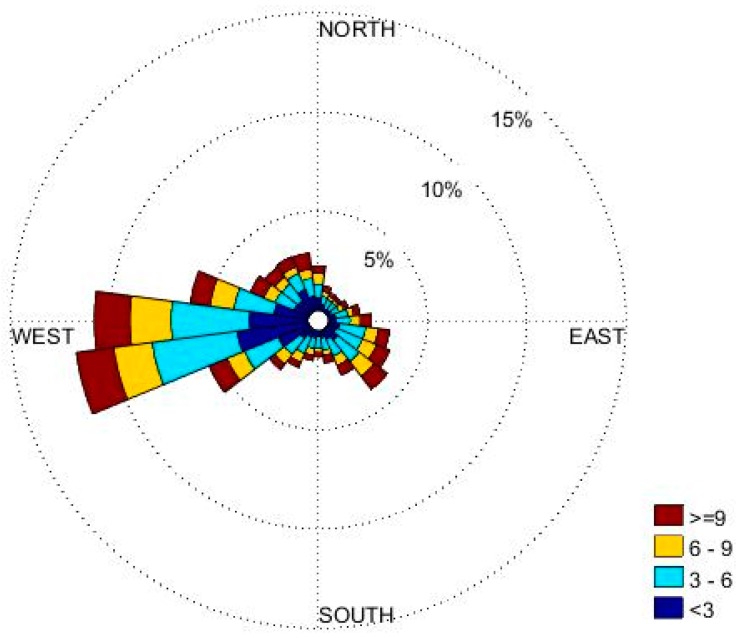
Wind rose of hourly winds from Imperial County Airport (KIPL) for 2017.

**Figure 5 ijerph-16-01252-f005:**
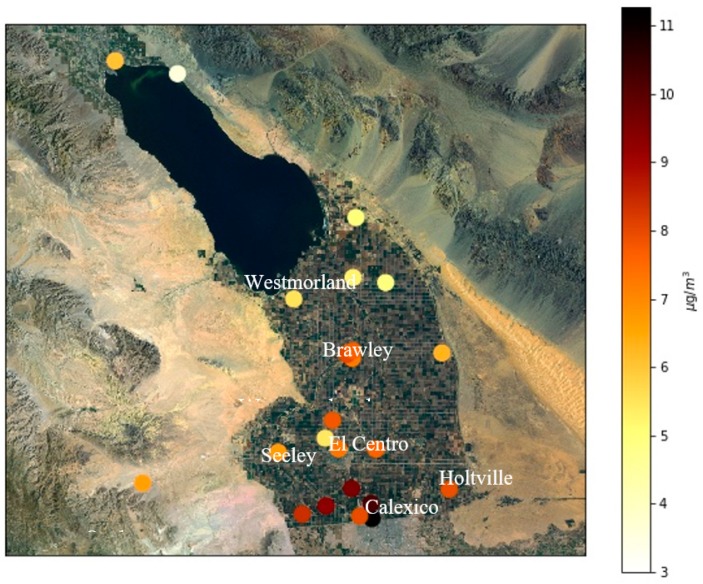
Annually average measured PM_2.5_ concentrations at selected sensors in the IVAN system for 2017.

**Figure 6 ijerph-16-01252-f006:**
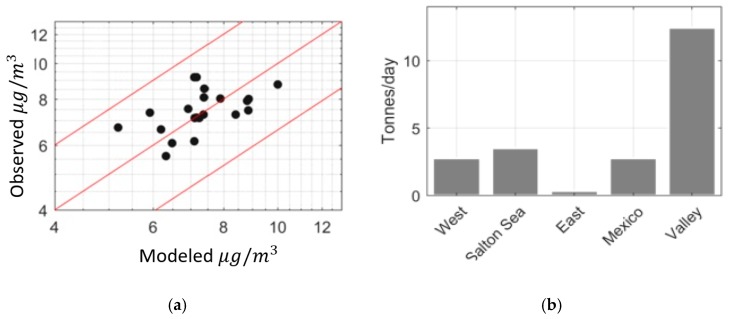
(**a**) Model performance for annually averaged data (red lines show factor of 1.5 deviations about observations). (**b**) Annual emission rates for different source categories inferred from the modeling.

**Figure 7 ijerph-16-01252-f007:**
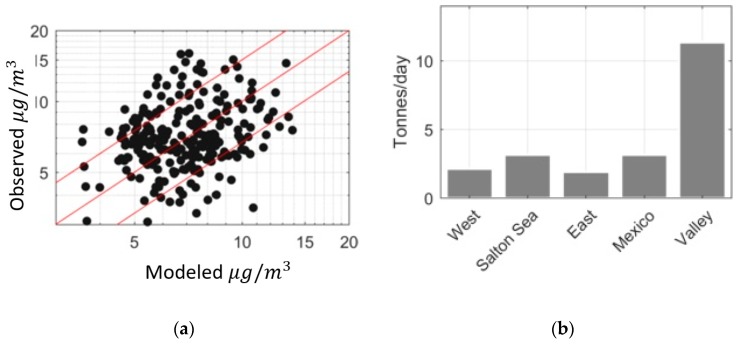
(**a**) Model performance for monthly averaged data (red lines show factor of 1.5 of observation). (**b**) Emission rates from different source categories.

**Figure 8 ijerph-16-01252-f008:**
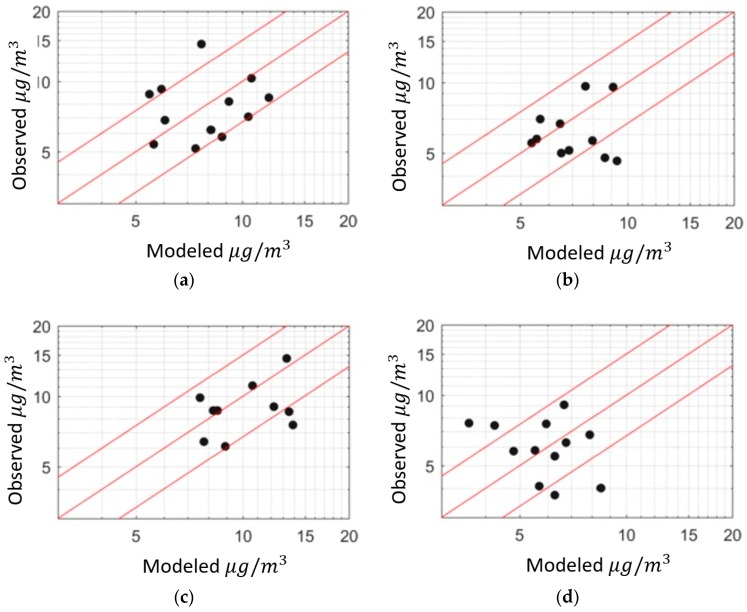
Model performance for different sensors for monthly averaged concentrations at (**a**) Brawley, (**b**) El Centro, (**c**) Calexico, and (**d**) Westmorland.

**Figure 9 ijerph-16-01252-f009:**
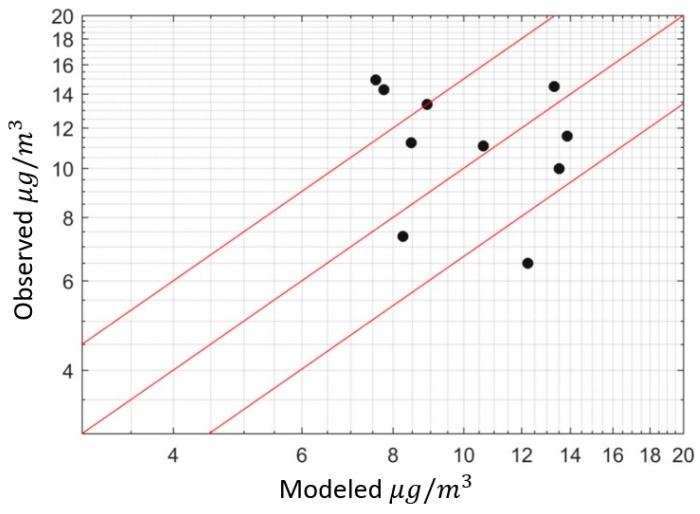
Monthly averaged model concentrations compared to measurements at the Calexico-Ethel St. regulatory station.

**Figure 10 ijerph-16-01252-f010:**
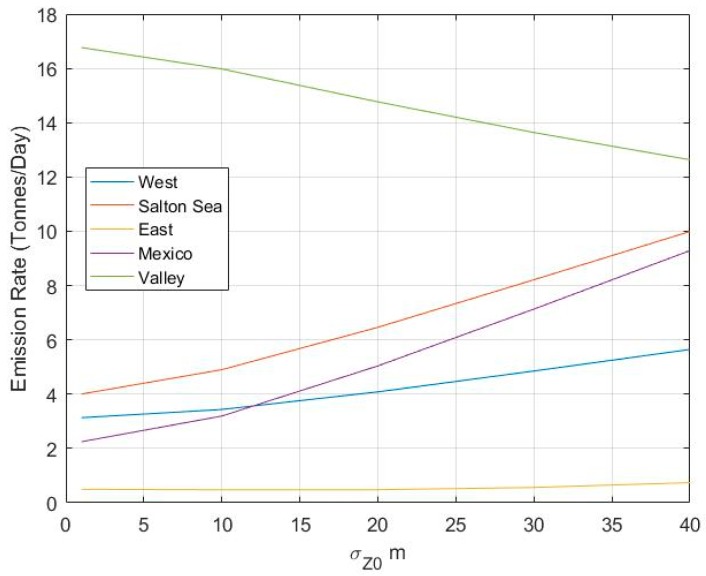
Model determined annual emission rates of different sources versus different prescribed inputs of initial plume spread (σ_z0_).

**Figure 11 ijerph-16-01252-f011:**
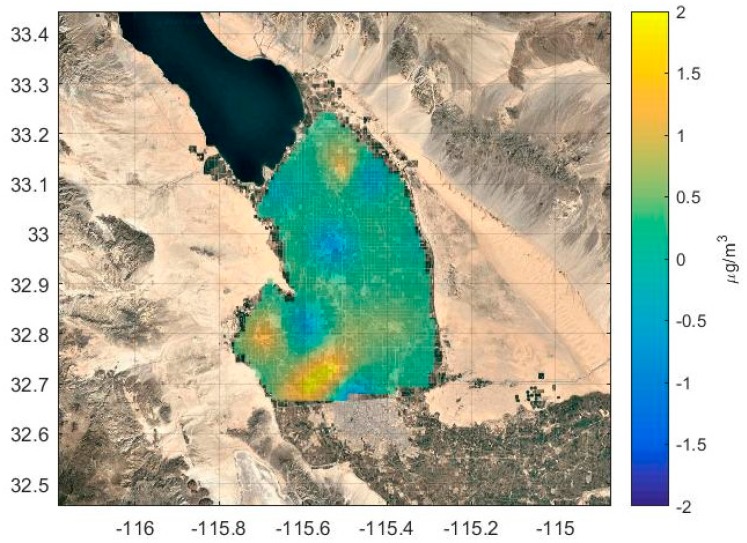
Map of residuals of annually averaged model minus measured concentrations. Map produced by Kriging residuals of model minus measured concentrations computed at IVAN receptors over the valley. See Section 3.3 for discussion.

**Figure 12 ijerph-16-01252-f012:**
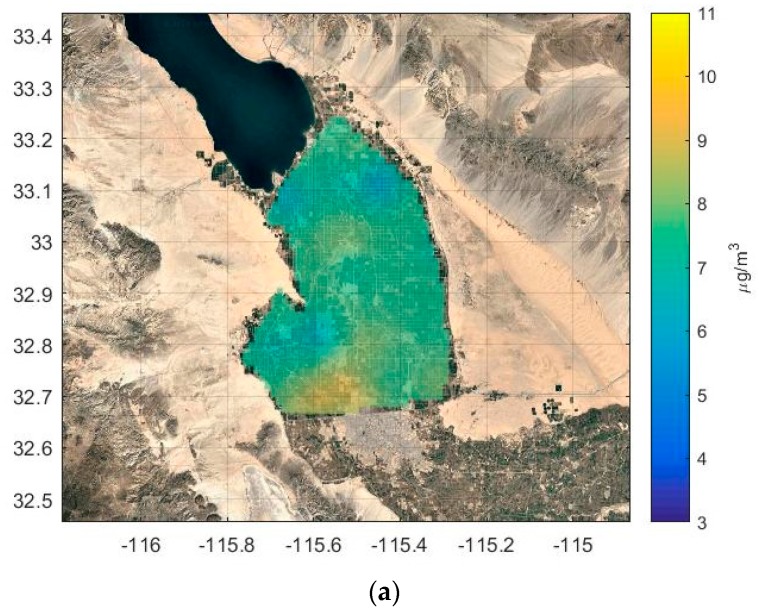
Annually averaged ${\mathrm{PM}}_{2.5}$ concentration maps for the year 2017 produced using (**a**) simple Kriging and (**b**) residual Kriging.

**Figure 13 ijerph-16-01252-f013:**
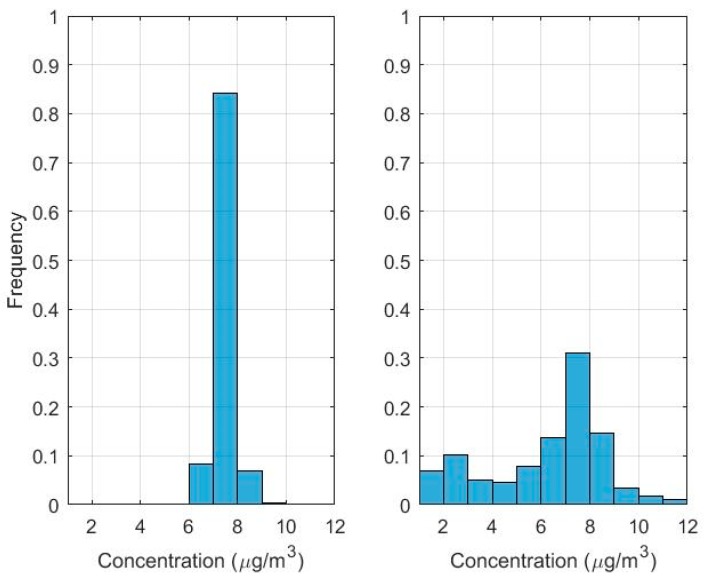
Histograms of PM_2.5_ concentrations over the 1000 m resolved receptors used to construct maps in Figure 12, calculated using (**a**) Simple Kriging, left panel, and (**b**) Residual Kriging, right panel.

**Figure 14 ijerph-16-01252-f014:**
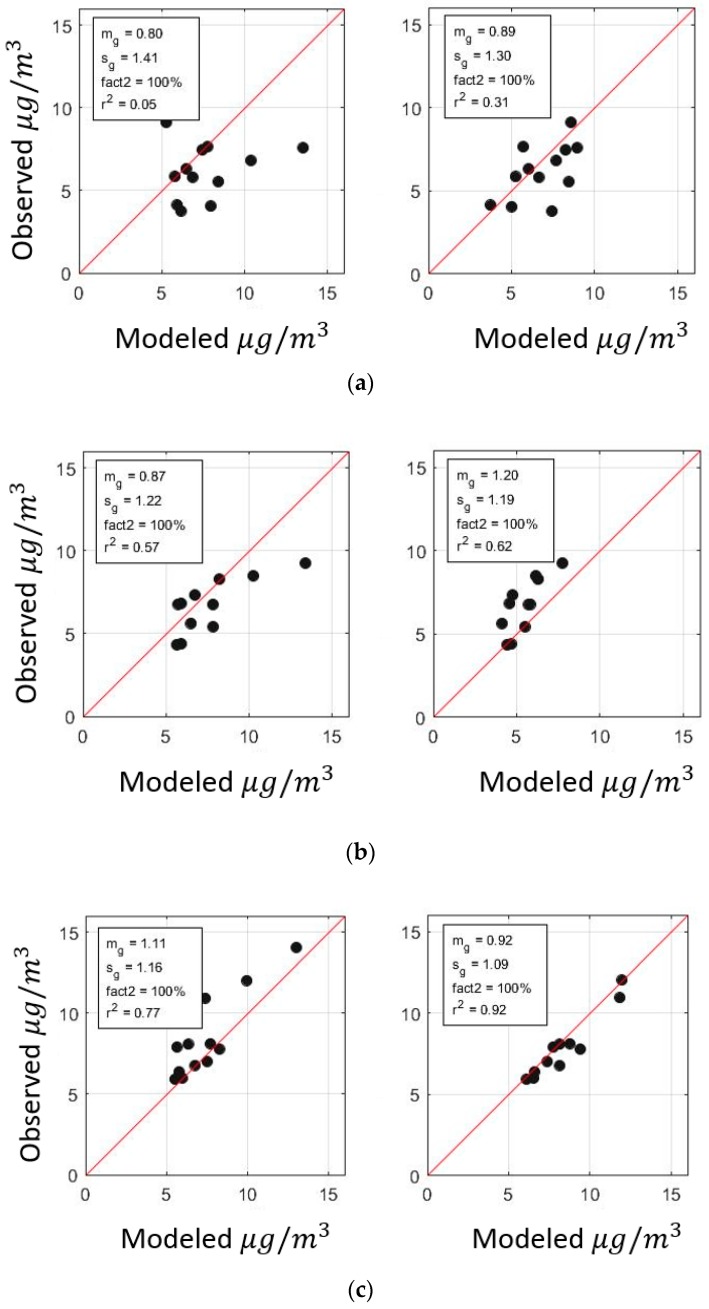
Cross-validation using simple Kriging model (left panels) and residual Kriging (right panels) at (**a**) Westmorland, (**b**) Seeley, and (**c**) Holtville.

**Table 1 ijerph-16-01252-t001:** Emission rates calculated by the model for annually averaged data.

Source	Emission Rates (Tons/Day)	95% Confidence Interval (Tons/Day)	95% Confidence Interval Range (Normalized to the Best Value)
West Desert	2.8	2.0–3.7	0.6
Salton Sea	3.5	1.5–5.8	1.2
East Desert	0.3	0.0–2.5	8.3
Mexico	2.8	1.6–4.1	0.9
Valley	12.4	9.0–15.3	0.5

**Table 2 ijerph-16-01252-t002:** Emission rates calculated by the model using monthly averaged data.

Source	Emission Rates (Tons/Day)	95% Confidence Interval (Tons/Day)	95% Confidence Interval Range (Normalized by the Best Value)
West Desert	2.1	1.5–2.8	0.6
Salton Sea	3.2	1.7–4.9	1.0
East Desert	1.9	0.6–3.3	1.4
Mexico	3.2	2.1–4.3	0.7
Valley	11.4	9.0–13.9	0.4

**Table 3 ijerph-16-01252-t003:** Emission rates and coefficient of determination of modeled versus measured monthly-averaged concentrations at 21 IVAN measurement stations for different months during 2017.

Month	Emission Rate (Tons per Day)	Coefficient of Determination (r2)
West Desert	Salton Sea	East Desert	Mexico	Valley
January	2.8	1.1	0	1.9	7.9	0.55
February	2.8	5.1	0	4.5	10.2	0.13
March	2.5	5.2	0	4.0	4.4	0.12
April	3.6	4.4	3.7	3.1	11.2	0.13
May	5.4	7.8	0.9	2.8	12.5	0.55
June	4.1	9.5	1.5	4.9	15.1	0.26
July	3.4	1.4	1.8	3.4	10.6	0.29
August	3.4	8.7	1.0	3.1	4.3	0.34
September	1.9	2.7	7.6	1.2	6.8	0.35
October	2.6	2.7	1.6	4.5	11.2	0.47
November	1.4	1.2	2.8	4.4	8.8	0.18
December	2.3	2.6	0	10.5	27.0	0.28
Average	3.0	4.3	1.7	4.0	10.8	
Standard Derivative	1.0	2.8	2.1	2.2	5.7

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
