# Peer review of "Using Low-Cost Air Quality Sensor Networks to Improve the Spatial and Temporal Resolution of Concentration Maps"

_ijerph, 2019, doi:10.3390/ijerph16071252_

Round 1

Reviewer 1 Report

This manuscript – “Using Low-Cost Air Quality Sensor Networks to Improve the Spatial and Temporal Resolution of Concentration Maps” describes data from low-cost air quality sensors in comparison to dispersion modelling results to obtain a finely resolved concentration map. Low-cost sensors have become more popular for air quality monitoring most recently. Such studies are needed to analyses the measurements of low-cost sensors, compare their measurements with reference sensors, and fuse their measurements with models predictions and measurements of reference sensors. However, before it can be accepted for publication, there are several issues with the current version of the manuscript which need addressing before it can be accepted for publication.  

1.       Line 54 – 59: provide reference. Some of these issues have been addressed in most recent literature. For example, LUR models developed more recently include meteorological data and road length or distance to main roads or other emission points and hence produce a dynamic maps of air pollution.

2.       Line 63: “to improve the spatial and temporal resolution of the LCAQM network,”, I think this is not true. The results does not show this. LCAQM can produce minute by minute hig resolution temporal data, which cannot be further improved by modelling outcome.

3.       Various sections need to be clearly identified (e.g., Methodology, Results etc.)

4.        Line 93:“The Dylos measurements are well correlated with regulatory ground station sensors”. How do you know this? Any evidence? Have you compared their measurements? Provide some example.

5.       What criterion was followed in the selection of sites for sensors deployment. In simple words, how you chose the 40 monitoring sites, where the sensors were deployed? Was the criterion based on population exposure, pollution concentrations, spatial variability of air pollution or anything else? Briefly describe the environment type of these sites.

6.       Give some example comparing PM2.5 concentrations measured by low-cost sensors with reference/accurate sensors. Also, describe the approach used for checking data quality of the low-cost sensors.

7.       Emission sources have been described vaguely. No values have been provided. Neither emission of each source are quantified. For example, how much PM2.5 was emitted by road traffic? How much PM2.5 came from roadside resuspension etc.

8.       The dispersion models employed require more description.  

9.       “Table 1. Emission rates calculated by the model for yearly averaged observation”. It is confusing – the word observation is used for measured concentrations not for the model prediction. Here both observation and estimation are mixed up, I think.

10.    Table 3. Correlation coefficient is r not r2. R2 is coefficient of determination.

11.    No comparison is made between modelled concentrations and observed concentrations from the reference sensors.  Also comparison should be made between low-cost sensors and reference sensors.

12.    Only comparison is made between modelled value and measurements of low-cost sensors, which are both questionable. How is the comparison justifiable?

13.     Some analysis (giving numbers) of the residual values should be included to mention at which receptor predicted value was higher than the observed value and vice versa.

Author Response

Comments and Suggestions for Authors

This manuscript – “Using Low-Cost Air Quality Sensor Networks to Improve the Spatial and Temporal Resolution of Concentration Maps” describes data from low-cost air quality sensors in comparison to dispersion modelling results to obtain a finely resolved concentration map. Low-cost sensors have become more popular for air quality monitoring most recently. Such studies are needed to analyses the measurements of low-cost sensors, compare their measurements with reference sensors, and fuse their measurements with models predictions and measurements of reference sensors. However, before it can be accepted for publication, there are several issues with the current version of the manuscript which need addressing before it can be accepted for publication.  

1.Line 54 – 59: provide reference. Some of these issues have been addressed in most recent literature. For example, LUR models developed more recently include meteorological data and road length or distance to main roads or other emission points and hence produce a dynamic maps of air pollution.

The Introduction was expanded to address this comment.

2.Line 63: “to improve the spatial and temporal resolution of the LCAQM network,”, I think this is not true. The results does not show this. LCAQM can produce minute by minute high resolution temporal data, which cannot be further improved by modelling outcome.

We agree. The introduction was modified to describe LUR models that yield temporally varying maps.

3.Various sections need to be clearly identified (e.g., Methodology, Results etc.)

The section headings were changed to address this.

4.Line 93:The Dylos measurements are well correlated with regulatory ground station sensors”. How do you know this? Any evidence? Have you compared their measurements? Provide some example.

We have included comparison of measurements from a  regulatory monitor with those from  the closest Dylos instrument.

5.What criterion was followed in the selection of sites for sensors deployment. In simple words, how you chose the 40 monitoring sites, where the sensors were deployed? Was the criterion based on population exposure, pollution concentrations, spatial variability of air pollution or anything else? Briefly describe the environment type of these sites.

We provide a description on how the locations of these monitors were chosen.

6.Give some example comparing PM2.5 concentrations measured by low-cost sensors with reference/accurate sensors. Also, describe the approach used for checking data quality of the low-cost sensors.

We provide an example of such a comparison.

7.Emission sources have been described vaguely. No values have been provided. Neither emission of each source are quantified. For example, how much PM2.5 was emitted by road traffic? How much PM2.5 came from roadside resuspension etc.

The modeling approach cannot identify specific sources. It only provides the total emissions from large areas such as the Imperial Valley, and from areas upwind of the boundaries of the region. In principle, we can obtain finer resolution of the emissions by using several source regions, but our approach cannot identify specific source types, such as resuspension and traffic.

8.The dispersion models employed require more description.  

We provide more details of the models, but the references are the primary source for the equations used in the model.

9.Table 1. Emission rates calculated by the model for yearly averaged observation”. It is confusing – the word observation is used for measured concentrations not for the model prediction. Here both observation and estimation are mixed up, I think.

The observation here means the yearly averaged data that was used in the calculation. The caption is changed to “Emission rates calculated by the model for yearly averaged data.”

10.Table 3. Correlation coefficient is r not r2. R2 is coefficient of determination.

Corrected.

11.No comparison is made between modelled concentrations and observed concentrations from the reference sensors. Also comparison should be made between low-cost sensors and reference sensors.

Comparison of low cost sensors and the regulatory station has been made in response to comment 4 with the only operating reference station near a sensor during the time of the data collection (Calexico station).

The performance of the model compared to the reference station was added as figure 9.

12.Only comparison is made between modelled value and measurements of low-cost sensors, which are both questionable. How is the comparison justifiable?

The validity of the model has been discussed as part of response to comment 4.

13.Some analysis (giving numbers) of the residual values should be included to mention at which receptor predicted value was higher than the observed value and vice versa.

A map of residuals has been added to the residual Kriging description to respond to this comment (figure 11), and we provide some plausible reasons for the observed distribution of residuals.

Reviewer 2 Report

In the paper, the authors describe an approach to analyzing PM2.5 data using Low-Cost Air Quality Monitors (LCAQM). 

The article presents an interesting topic and is not only well structured and written but also easy to follow. In general, the technical description is acceptable both in terms of depth and range. However, I have some comments as follows.

In one hand, the LCAQM specification must be succinctly presented in the paper, particularly,  the accuracy and sensitivity. On the other hand, there is a lot of similar LCAQM  published recently.  

Johnston, S.J.; Basford, P.J.; Bulot, F.M.J.; Apetroaie-Cristea, M.; Easton, N.H.C.; Davenport, C.; Foster, G.L.; Loxham, M.; Morris, A.K.R.; Cox, S.J. City Scale Particulate Matter Monitoring Using LoRaWAN Based Air Quality IoT Devices. Sensors 2019, 19, 209.

Marques, G.; Roque Ferreira, C.; Pitarma, R. A System Based on the Internet of Things for Real-Time Particle Monitoring in Buildings. Int. J. Environ. Res. Public Health 2018, 15, 821.

Fazio, E.; Bonacquisti, V.; Di Michele, M.; Frasca, F.; Chianese, A.; Siani, A.M. CleAir Monitoring System for Particulate Matter: A Case in the Napoleonic Museum in Rome. Sensors 2017, 17, 2076.

The LCAQM used in the tests must be succinctly compared with existing ones to justify the findings of this study.

 The paper must have a specific section describing the related studies and their shortcomings. The proposal should describe the novelties on existing proposals based on scientific evidence.

The authors should also improve the conclusions section with more information about the further steps for this field and the limitations of the work done. 

Author Response

Comments and Suggestions for Authors

In the paper, the authors describe an approach to analyzing PM2.5 data using Low-Cost Air Quality Monitors (LCAQM). 

The article presents an interesting topic and is not only well structured and written but also easy to follow. In general, the technical description is acceptable both in terms of depth and range. However, I have some comments as follows.

1.     In one hand, the LCAQM specification must be succinctly presented in the paper, particularly,  the accuracy and sensitivity.

We provide a comparison of measurements from a LCAQM with those from the closet regulatory monitor.

2. On the other hand, there is a lot of similar LCAQM  published recently.  

Johnston, S.J.; Basford, P.J.; Bulot, F.M.J.; Apetroaie-Cristea, M.; Easton, N.H.C.; Davenport, C.; Foster, G.L.; Loxham, M.; Morris, A.K.R.; Cox, S.J. City Scale Particulate Matter Monitoring Using LoRaWAN Based Air Quality IoT Devices. Sensors 2019, 19, 209.

Marques, G.; Roque Ferreira, C.; Pitarma, R. A System Based on the Internet of Things for Real-Time Particle Monitoring in Buildings. Int. J. Environ. Res. Public Health 2018, 15, 821.

Fazio, E.; Bonacquisti, V.; Di Michele, M.; Frasca, F.; Chianese, A.; Siani, A.M. CleAir Monitoring System for Particulate Matter: A Case in the Napoleonic Museum in Rome. Sensors 2017, 17, 2076.

The LCAQM used in the tests must be succinctly compared with existing ones to justify the findings of this study.

The Introduction was expanded to address this comment, and the required comparison is presented in the paper.

4.The paper must have a specific section describing the related studies and their shortcomings. The proposal should describe the novelties on existing proposals based on scientific evidence.

The Introduction was expanded to address this comment.

5.The authors should also improve the conclusions section with more information about the further steps for this field and the limitations of the work done. 

A section on future steps was added to the conclusions.

Round 2

Reviewer 2 Report

The authors accepted the suggested comments and improved the quality of the paper.